# Redefining player roles in professional women's basketball: From traditional positions to functional profiles

Sergio J. Ibáñez[1], Javier Courel-Ibáñez[2]*, José Miguel Contreras-García[3], María Isabel Piñar-López[4]

**1** Faculty of Sport Sciences, Training Optimization and Sports Performance Research Group (GOERD), University of Extremadura, Cáceres, Spain, **2** Department of Physical Education and Sports, Faculty of Education and Sport Sciences, University of Granada, Melilla, Spain, **3** Department of Didactics of Mathematics, University of Granada, Granada, Spain, **4** Department of Physical Education and Sports, Faculty of Sport Sciences, University of Granada, Granada, Spain

\* courel@ugr.es

**Data availability statement:** All data analyzed in this study are publicly available through the official website of Liga Femenina Endesa (https://www.lfendesa.es/inicio.aspx). Processed data and analytical scripts can be provided by the corresponding author upon reasonable request.

## Abstract

The analysis of box-score performance indicators has traditionally been used to classify player roles in women's basketball based on the five conventional positions: point guard, shooting guard, small forward, power forward, and center. However, this framework may not reflect the current tactical and functional demands of the game. The aim of this study was to identify and redefine functional player roles in professional women's basketball using performance data derived from actual competition. A total of 36,204 individual player records from 3,894 games in the Spanish Liga Femenina Endesa (2012–2022) were analyzed. Game-related statistics were normalized by effective playing time and scaled to a 40-minute format. One-way ANOVA revealed significant differences across traditional positions, but also indicated considerable functional overlap. Unsupervised learning techniques, including k-means and Gaussian mixture models, were applied to identify underlying performance-based player profiles. The analysis yielded nine stable and interpretable functional roles, offering a more nuanced classification than the traditional five-position model. These roles capture offensive, defensive, and hybrid specializations, providing coaches and analysts with a practical framework for tactical planning, scouting, and individualized player development. The findings support a shift toward data-driven classification systems that better reflect the functional realities of modern elite women's basketball.

## 1. Introduction

Basketball was originally designed as a structured indoor sport and initially featured teams of nine players without defined tactical roles. As the game developed, the number of players was standardized to five per team by 1893, creating a framework

**Funding:** This research was partially funded by the Research Group Support Grant (GR24133). It was co-funded at 85% by the European Union through the European Regional Development Funds (ERDF), and by the Regional Government of Extremadura (Department of Education, Science, and Vocational Training). The Managing Authority is the Ministry of Finance of Spain. JCI is supported by the Unit of Research Excellence of the University of Granada, Melilla Campus, UECUMel (UCE-PP2024-02). The funders had no role in study design, data collection and analysis, decision to publish, or preparation of the manuscript.

**Competing interests:** The authors have declared that no competing interests exist.

that enabled the emergence of positional specialization. This transition supported the differentiation of three foundational roles: guards, primarily responsible for defense; forwards, focused on scoring; and centers, contributing to both offensive and defensive actions. These tactical functions marked the beginning of role-based structure in team strategy and player deployment [1].

Although this tripartite classification remains nominally intact, the demands of modern basketball have led to a significant transformation in role execution. The conventional five-position system—Point Guard (PG), Shooting Guard (SG), Small Forward (SF), Power Forward (PF), and Center (C)—emerged to reflect more specialized functions [2]. However, the increasing tactical complexity and fluidity of the game have challenged the adequacy of this model, giving rise to the need for more nuanced classifications based on actual in-game behavior [3].

To address this shift, researchers have increasingly applied multivariate statistical techniques to redefine basketball roles. One of the earliest efforts proposed a system of 13 functional roles based on players' statistical profiles rather than nominal positions, offering a radically different framework for understanding performance [4,5]. Subsequent work confirmed the limitations of the traditional model. For instance, player clustering based on per-minute contributions revealed seven distinct profiles [6], while analysis of key in-game actions suggested a four-role structure [7]. Other classifications have identified five roles grouped into three performance clusters [8], or defined by discrete tactical behaviors and efficiency metrics [9].

Expanding on this body of work, a recent study spanning two decades of NBA data used hierarchical clustering and dimensionality reduction to generate role taxonomies, illustrating the value of visual analytics in performance profiling [10]. Using a different methodological approach, weighted network clustering has also yielded eight archetypal roles, each offering greater explanatory power for team performance than traditional labels [11]. These contributions converge on a clear conclusion: static positional models fail to capture the complexity of contemporary basketball.

Most existing research maintains traditional positional structures, despite growing evidence of performance variation that challenges their validity. A large-scale analysis of elite competition identified four functional variants within PG, SG, and SF roles, five within PF, and six within the C position—highlighting the inadequacy of nominal roles to reflect actual game responsibilities [2]. In men's basketball, the study of positional performance has also incorporated contextual variables such as game location [12], the relative age effect [13], and career development trajectories [14]. Efforts to profile functional roles have extended to youth and developmental levels, where player clustering has been linked to both competitive context [15] and team success [16,17]. In contrast, women's basketball has been slower to adopt function-based classification models. Analyses of elite-level competition, including the WNBA and Spain's Liga Endesa, have revealed distinct contributions by position—for example, higher assist rates among guards and greater offensive rebounding among interior players [18–20]. Yet these findings remain anchored in static definitions.

Recent research has argued that such rigidity obscures meaningful tactical differences. One study of players in continental competitions reported substantial physical

and strategic heterogeneity within positional groups, underscoring the limitations of current models [21]. Another analysis of Euroleague Women players confirmed that while position remains statistically relevant, it does not adequately account for observed performance variation [22].

Compared to men's basketball, where advanced methods have produced dynamic, performance-based profiles, the women's game continues to rely on traditional designations. This limits insight into key elements such as versatility, tactical role differentiation, lineup optimization, and player development planning. There is a clear gap in the literature concerning functionally meaningful classification in elite women's basketball.

The present study seeks to address this gap by pursuing two objectives: (i) to describe and identify the performance indicators that differentiate players according to their traditional specific position (PG, SG, SF, PF, and C), and (ii) to propose a new functional classification, derived from performance indicators, that more accurately reflects actual in-game roles in professional women's basketball.

## 2. Materials and methods

### 2.1. Research design

This study adopted a quantitative, observational methodology consistent with performance and notational analysis frameworks [23,24]. An ex post facto, descriptive–comparative design was used to examine objective performance data obtained from official match records, without experimental manipulation. The analysis focused on identifying functional performance patterns in professional women's basketball players by comparing in-game actions across traditional positional roles.

### 2.2. Sample

The sample consisted of 36,204 individual player records from 3,894 official games of the Liga Femenina Endesa, spanning the 2012–2013–2021–2022 seasons. The dataset included regular season games, playoff matches (quarterfinals and semifinals), Supercopa (semifinal and final), and Copa de la Reina (quarterfinals, semifinals, and final). All player participation was considered, accounting for team roster changes throughout the ten seasons.

### 2.3. Data collection

An ad hoc data recording sheet was specifically designed for this study to collect and organize the selected variables according to the research objectives. This sheet structured information by player, match, and season, ensuring traceability and consistency across records. Data were sourced from the official Liga Femenina Endesa website (https://www.lfendesa.es/inicio.aspx), a publicly accessible platform that provides validated and regularly updated information on national women's basketball competitions in Spain.

### 2.4. Variables

The independent variable was the player's traditional positional role (PG, SG, SF, PF, and C). This positional classification served as a criterion to explore differential behaviors based on in-game actions, and as a foundation for the functional reclassification proposed in this study.

Dependent variables included game indicators extracted from the official box score. Although box-score statistics are limited to capturing off-ball actions, spatial dynamics, or advanced defensive behaviors, these variables quantify each player's contributions across various aspects of the game and were selected based on their tactical relevance and representation of individual and team performance. The variables considered were: minutes played, field goals made and attempted (2-point, 3-point, and free throws), offensive and defensive rebounds, assists, steals, turnovers, blocks for and against, fouls committed and received.

## 2.5. Statistical analyses

All performance indicators were normalized by actual playing time to account for differences in player exposure, calculated as per-minute values and then scaled to a standard 40-minute reference, consistent with official FIBA match duration [14,21,25] Descriptive statistics (means and standard deviations) were used to characterize player performance across traditional positions.

To explore functional profiles beyond nominal roles, an unsupervised clustering approach was employed. Initial efforts to determine the optimal number of clusters used standard metrics such as the elbow method, silhouette index, and gap statistic [26]. Clustering was conducted using both *k*-means and model-based Gaussian mixture models (GMM) via the *nclust* package, yielding stable and meaningful groupings. The k-means algorithm was run for 10 iterations and used Euclidean distance, automatic procedures used by default, following previous methodologies in similar sports contexts [8]. For the application of Gaussian mixture models (GMM), it was assumed that the data of each cluster followed a multivariate normal distribution. The silhouette index was explicitly incorporated as a complementary criterion for selecting the optimal number of clusters [27]. These clusters reflected distinct playing styles in terms of offensive efficiency, rebounding, usage, distribution, and shooting tendencies. The final selection of nine clusters produced reasonably balanced group sizes and facilitated a typology aligned with real game demands.

Following cluster generation, functional group labels were established through a Delphi method involving four PhD-level experts with published work in performance analysis and more than 10 coaching experience in both men's and women's basketball at national or professional level [28]. This panel evaluated the cluster characteristics and proposed nominal definitions based on tactical interpretation.

To assess differences between traditional positions and the newly identified functional clusters, a one-way analysis of variance (ANOVA) was conducted. Bonferroni-adjusted post hoc tests were used to identify pairwise differences among groups [29]. In addition, paired samples t-tests were used to evaluate the statistical impact of time normalization on each performance indicator, comparing raw means to normalized values using the formula [30]:

$$Difference\ \% = ((Normalized\ Value - Mean)/Mean) \times 100$$

All statistical analyses were performed using IBM SPSS Statistics v27. Data organization and visualization were conducted using Python (v3.11), with the *pandas* library for data management and *matplotlib* and *seaborn* for graphical representation.

## 3. Results

### 3.1. Game indicators by traditional playing position

Fig 1 presents the descriptive statistics for each of the five traditional playing positions in professional women's basketball, including both raw and normalized values per 40 minutes of effective playing time. Time normalization resulted in a general increase in performance values across all positions, particularly for volume-dependent indicators such as shooting attempts and rebounds. This adjustment allowed for more equitable comparisons of player productivity, controlling for unequal playing time distributions.

Notable positional differences were evident in the functional profiles. SFs displayed a relatively balanced contribution across offensive and defensive domains, while other roles demonstrated clearer specialization- PGs exhibited the highest values in ball distribution and defensive pressure, with a mean of 2.24 assists and 1.05 steals per 40 minutes. In contrast, Cs dominated in rim protection and rebounding, leading in offensive rebounds (1.59), defensive rebounds (3.43), and blocks (0.39). SGs recorded the highest three-point shooting volume, while PFs combined interior scoring and rebounding contributions, reflecting a hybrid performance profile.

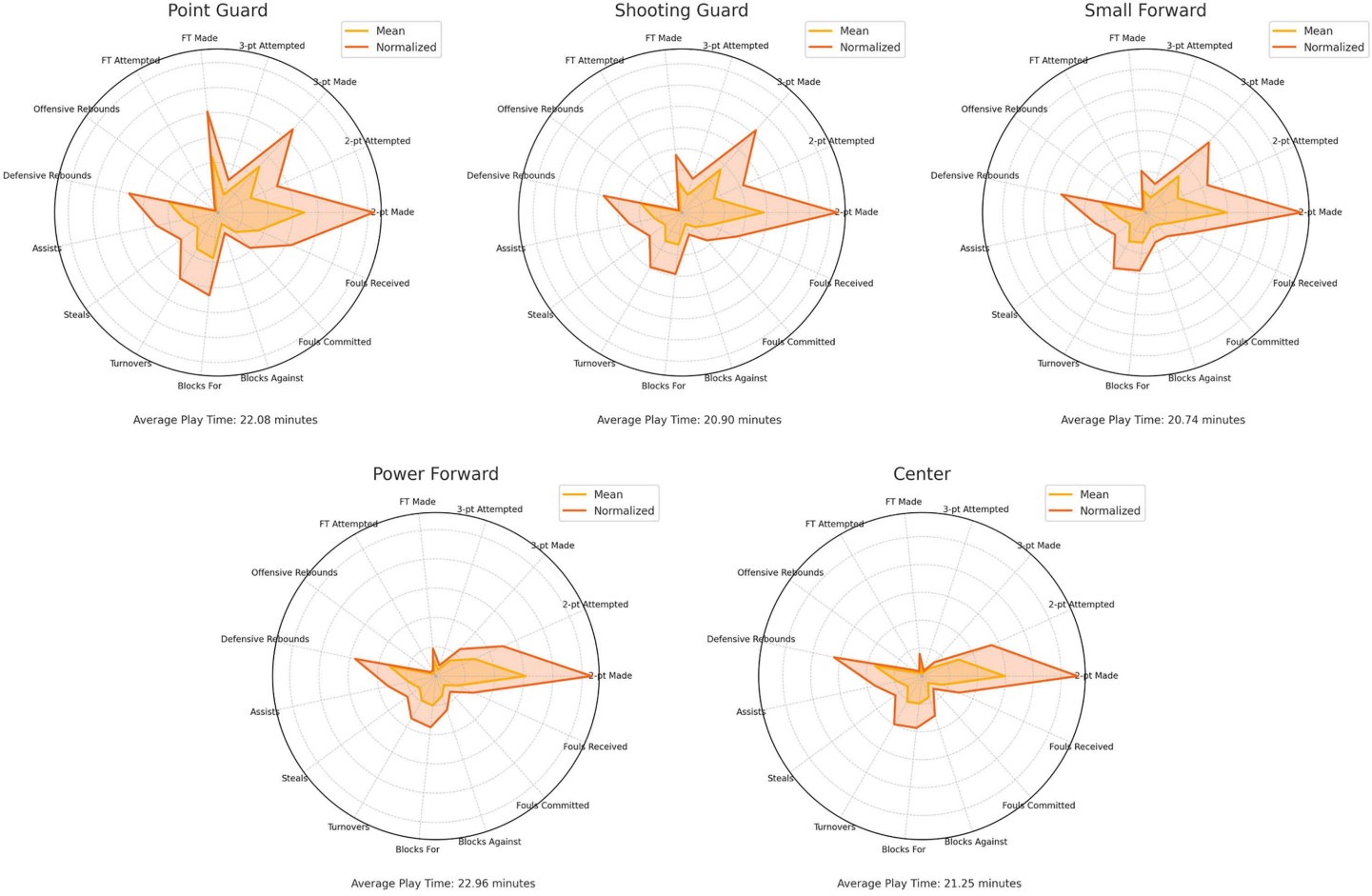

**Fig 1. Radar plots of positional performance profiles based on raw and normalized metrics.** Each panel represents one of the five basketball playing positions (Point Guard, Shooting Guard, Small Forward, Power Forward, and Center). Orange lines display the average raw metric values, while red lines represent the normalized values used in subsequent analyses. Metrics include various scoring, rebounding, passing, and defensive indicators.

A one-way analysis of variance (ANOVA) revealed significant differences across all performance indicators by playing position ($p < .001$). Post hoc Bonferroni tests confirmed statistically significant pairwise differences between most positional groupings (Table 1), supporting the presence of distinct role-based performance patterns. However, the overlapping ranges observed in normalized metrics also suggest an increasing degree of functional convergence between perimeter and interior roles, consistent with trends toward positional fluidity in modern basketball.

### 3.2. Comparison between raw and normalized values

Normalization resulted in substantial percentage increases across indicators, with average boosts ranging from 60% to 100%, particularly in volume-based metrics such as shot attempts, made field goals, and rebounds (Fig 2). Paired samples t-tests confirmed that all differences between raw and normalized values were statistically significant ($p < 0.010$), indicating that temporal normalization has a systematic impact on performance interpretation. This effect was especially pronounced in positions typically characterized by lower average playing time, such as PGs and SGs. Distributional

**Table 1. Differences in raw and normalized per-40-minute performance metrics across traditional playing positions.**

| Performance indicator | Point Guard | | Shooting Guard | | Shooting Forward | | Power Forward | | Center | | p-value |
|---|---|---|---|---|---|---|---|---|---|---|---|
| | M (SD) | 40-min | M (SD) | 40-min | M (SD) | 40-min | M (SD) | 40-min | M (SD) | 40-min | |
| 2P Field Goals Made | 1.4 (1.6) | 2.6 | 1.7 (1.9) | 3.2 | 1.7 (1.9) | 3.3 | 2.9 (2.5) | 5.0 | 2.9 (2.4) | 5.5 | <0.001 |
| 2P Field Goals Attempt | 3.4 (3.1) | 6.2 | 3.8 (3.4) | 7.3 | 4.0 (3.4) | 7.6 | 6.1 (4.3) | 10.6 | 5.9 (4.0) | 11.2 | <0.001 |
| 3P Field Goals Made | 0.8 (1.0) | 1.4 | 0.9 (1.2) | 1.7 | 0.8 (1.1) | 1.5 | 0.4 (0.8) | 0.8 | 0.2 (0.6) | 0.4 | <0.001 |
| 3P Field Goals Attempt | 2.5 (2.1) | 4.5 | 2.7 (2.4) | 5.2 | 2.4 (2.2) | 4.6 | 1.4 (1.8) | 2.5 | 0.7 (1.3) | 1.3 | <0.001 |
| Free Throws Made | 1.0 (1.6) | 1.8 | 1.0 (1.6) | 1.9 | 1 (1.5) | 1.9 | 1.4 (1.8) | 2.4 | 1.3 (1.7) | 2.4 | <0.001 |
| Free Throws Attempt | 1.4 (2.0) | 2.5 | 1.3 (2.0) | 2.5 | 1.3 (1.9) | 2.5 | 1.9 (2.3) | 3.3 | 1.8 (2.2) | 3.4 | <0.001 |
| Offensive Rebounds | 0.5 (0.8) | 0.9 | 0.6 (0.9) | 1.1 | 0.8 (1.1) | 1.5 | 1.4 (1.5) | 2.4 | 1.6 (1.7) | 3.0 | <0.001 |
| Defensive Rebounds | 2.0 (1.9) | 3.6 | 2.0 (1.9) | 3.8 | 2.2 (2.0) | 4.2 | 3.2 (2.6) | 5.6 | 3.4 (2.8) | 6.5 | <0.001 |
| Assists | 2.2 (2.1) | 4.1 | 1.4 (1.6) | 2.7 | 1.1 (1.3) | 2.0 | 1.1 (1.3) | 1.9 | 0.9 (1.1) | 1.6 | <0.001 |
| Steals | 1.1 (1.2) | 1.9 | 0.9 (1.2) | 1.8 | 0.8 (1.0) | 1.6 | 0.8 (1.1) | 1.5 | 0.7 (0.9) | 1.2 | <0.001 |
| Turnovers | 1.8 (1.6) | 3.2 | 1.5 (1.5) | 2.8 | 1.3 (1.3) | 2.5 | 1.6 (1.5) | 2.8 | 1.6 (1.5) | 2.9 | <0.001 |
| Blocks For | 0.1 (0.3) | 0.1 | 0.1 (0.3) | 0.2 | 0.1 (0.4) | 0.3 | 0.3 (0.6) | 0.5 | 0.4 (0.7) | 0.7 | <0.001 |
| Blocks Against | 0.2 (0.4) | 0.3 | 0.2 (0.4) | 0.3 | 0.2 (0.4) | 0.3 | 0.2 (0.6) | 0.4 | 0.2 (0.5) | 0.4 | <0.001 |
| Fouls Committed | 1.7 (1.3) | 3.0 | 1.6 (1.3) | 3.0 | 1.6 (1.3) | 3.1 | 1.9 (1.4) | 3.3 | 2.1 (1.4) | 4.0 | <0.001 |
| Fouls Received | 1.8 (1.8) | 3.3 | 1.5 (1.7) | 2.9 | 1.5 (1.6) | 2.9 | 2.0 (1.9) | 3.5 | 2.0 (1.8) | 3.8 | <0.001 |

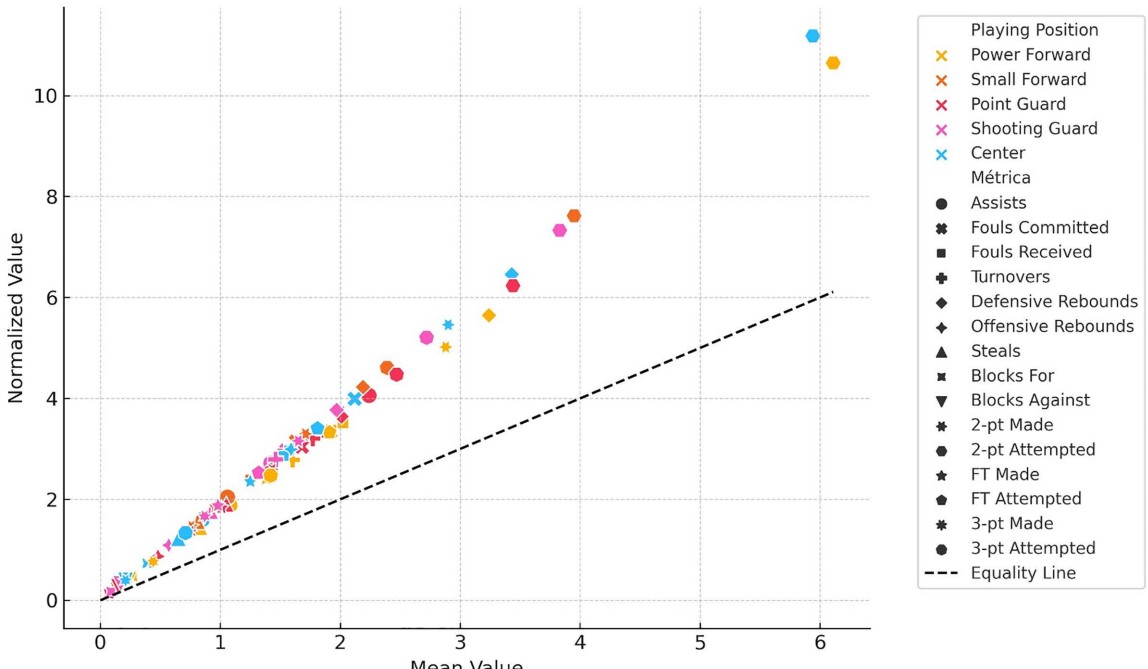

**Fig 2. Relationship between raw mean values and normalized values of performance metrics across playing positions.** Each point represents a specific basketball performance indicator, with shape indicating the metric and color representing the player's position. The x-axis shows the raw mean value of each indicator, while the y-axis displays its normalized (standardized) value used for clustering and classification analysis. The dashed diagonal line represents equality (i.e., points where the normalized value equals the raw mean), serving as a visual reference for relative scaling across metrics.

analyses showed greater variability in these perimeter roles, suggesting that relying solely on raw statistics may lead to underestimation of their actual contributions.

### 3.3. Functional classification through cluster analysis

The analysis of data from ten full seasons of Spain's top-tier women's basketball league enabled the establishment of a functional classification of players based on individual and collective performance indicators. The silhouette index graph showed a clear initial inflection point at $k=2$, suggesting a basic structure of two large player groups based on playing characteristics (Fig 3). While this solution is statistically efficient in terms of reducing within-group variance, it was deemed practically inadequate, as it simply distinguished between higher- and lower-performing players. A second inflection point appeared at $k=4$, which corresponds to the number of current traditional positions. However, further analysis of the curve revealed a tendency toward stabilization between five and nine clusters. After applying the model-based Gaussian mixture clustering, a stable and conceptually interpretable solution emerged at $k=9$ (Fig 3). This classification offers greater precision in reflecting the functional diversity observed in today's game and aligns with the practical needs of coaching and technical staff in professional women's basketball. These professionals require more differentiated profiles to optimize tactical planning, the design of personalized training, and strategic recruitment efforts.

### 3.4. Cluster distribution and positional composition

Analysis of the distribution of players across the nine functional clusters revealed substantial heterogeneity in group sizes (Fig 4). Cluster 5 represented the largest subgroup, encompassing approximately 25% of the total sample, followed by

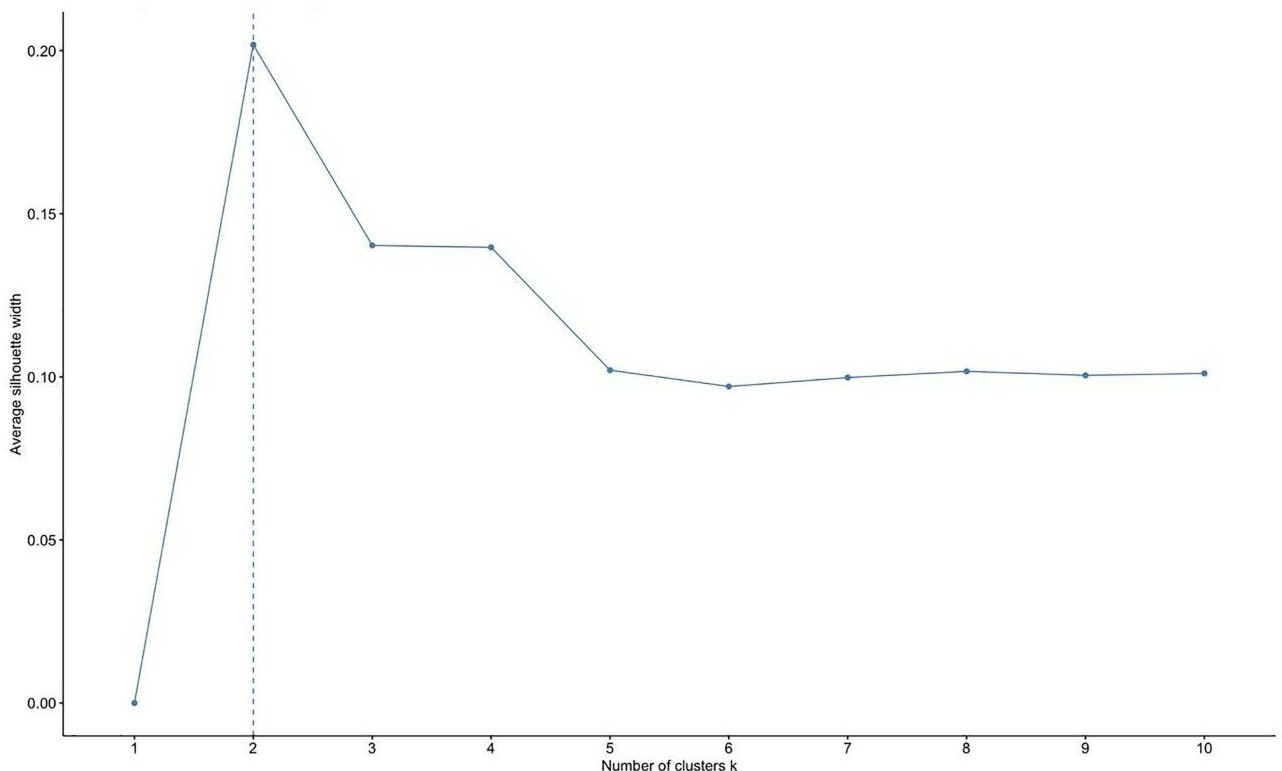

**Fig 3. Optimal number of clusters according to average silhouette width.**

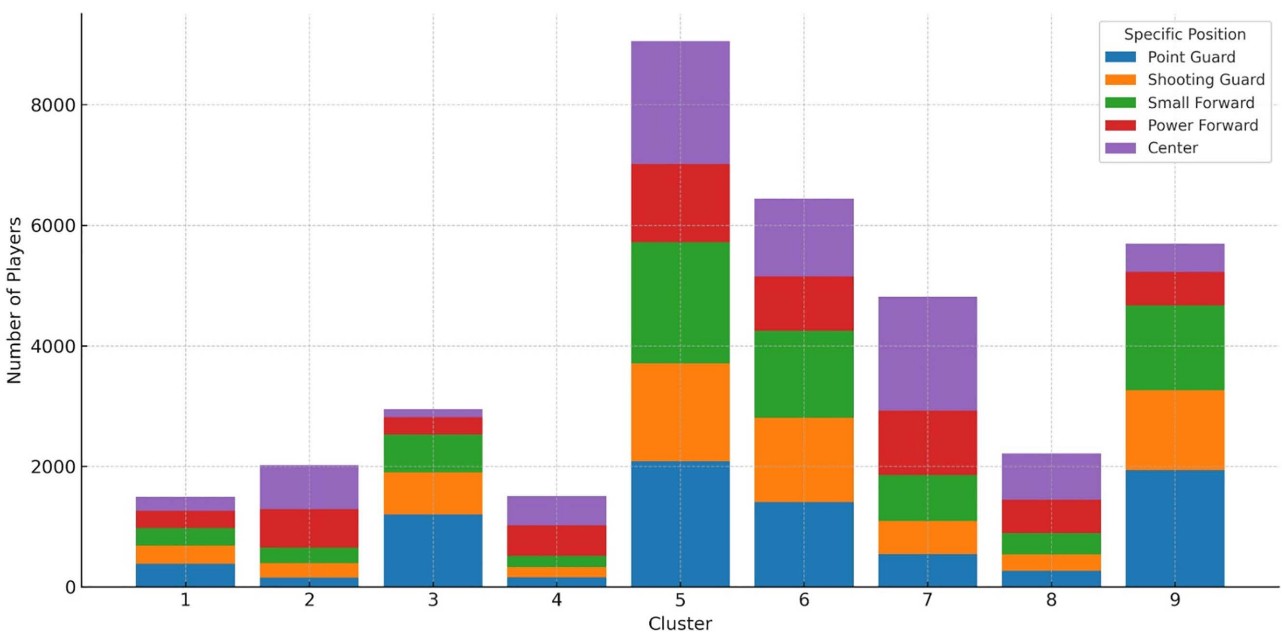

**Fig 4. Distribution of playing positions across the nine identified clusters.** Stacked bar chart showing the number of players in each cluster, broken down by specific playing positions (Point Guard, Shooting Guard, Small Forward, Power Forward, and Center). The height of each bar represents the total number of players assigned to a given cluster, while the color segments within each bar indicate the positional composition.

Cluster 6 (17.8%) and Cluster 9 (15.7%). In contrast, Clusters 1, 2, 4, and 8 accounted for less than 6% each, indicating a more specialized or less frequent performance profile. Several clusters demonstrated a clear positional dominance. For instance, Cluster 2 (interpreted as "Skilled Forward"), Cluster 4 ("Ball-Dominant Scorer"), and Cluster 7 ("Mid-Range Big") were primarily composed of PFs and Cs, comprising 67.52%, 65.89%, and 61.40% of their respective groups. Conversely, Cluster 3 ("Floor General") and Cluster 9 ("Three-Point Shooting Guard") were predominantly formed by perimeter players—PGs, SGs, and small forwards—accounting for 85.5% and 81.95%, respectively.

It is noteworthy that Clusters 5 and 6, which together included 43.71% of all players, were characterized by the lowest average playing times (16.38 and 6.05 minutes per game, respectively). These were interpreted as second-rotation profiles. The remaining clusters were distributed among first-rotation players (averaging 20–25 minutes) and starters (five profiles with average playing time exceeding 30 minutes), suggesting a functional typology aligned with role hierarchy and in-game utilization.

### 3.5. Player classification into nine clusters based on performance indicators (raw and normalized)

One-way ANOVA confirmed significant differences across clusters ($p < 0.001$), validating the segmentation and supporting the interpretative value of the identified profiles (Table 2). Compared to the conventional five-position model, this data-driven classification enabled a more refined differentiation, capturing hybrid and specialized player roles not adequately represented by traditional frameworks.

Each cluster reflected a unique performance pattern. For example, Cluster 4 comprised high-involvement players with elevated values in two-point field goals (attempted and made), rebounds (offensive and defensive), and overall minutes played—suggesting a central offensive and rebounding role. In contrast, Cluster 6 included low-participation players with uniformly low output across all indicators, characteristic of second-rotation roles. Cluster 3 exhibited high values in both three-point shooting and assists, indicating a versatile perimeter role with playmaking and scoring responsibilities.

**Table 2. Differences in raw performance metrics across the 9 clusters.**

| Performance indicator | Cluster 1 | Cluster 2 | Cluster 3 | Cluster 4 | Cluster 5 | Cluster 6 | Cluster 7 | Cluster 8 | Cluster 9 | p-value |
|---|---|---|---|---|---|---|---|---|---|---|
| Minutes Played | 32.5 (3.5) | 30.2 (2.8) | 34.7 (2.8) | 37.6 (2.7) | 16.4 (2.7) | 6.1 (3.3) | 22.7 (2.4) | 30.6 (2.3) | 25.9 (2.5) | <0.001 |
| 2P Field Goals Made | 3.1 (1.7) | 6.1 (2.0) | 2.2 (1.5) | 6.3 (2.2) | 1.1 (1.2) | 0.3 (0.6) | 3.4 (1.7) | 3.2 (1.4) | 1.3 (1.1) | <0.001 |
| 2P Field Goals Attempt | 6.8 (2.5) | 11.9 (2.3) | 5.2 (2.4) | 12.8 (3.2) | 2.8 (1.9) | 0.9 (1.1) | 6.9 (2.2) | 7.1 (1.9) | 3.3 (1.8) | <0.001 |
| 3P Field Goals Made | 1.0 (1.2) | 0.4 (0.8) | 1.7 (1.4) | 0.6 (1.0) | 0.4 (0.8) | 0.1 (0.3) | 0.3 (0.7) | 0.4 (0.8) | 1.1 (1.2) | <0.001 |
| 3P Field Goals Attempt | 3.3 (2.5) | 1.5 (1.7) | 4.8 (2.6) | 2.0 (2.3) | 1.5 (1.6) | 0.5 (0.8) | 1.2 (1.4) | 1.5 (1.6) | 3.4 (2.1) | <0.001 |
| Free Throws Made | 5.2 (1.8) | 1.7 (1.5) | 1.1 (1.2) | 3.2 (2.2) | 0.6 (1.0) | 0.2 (0.6) | 1.6 (1.6) | 1.3 (1.3) | 0.8 (1.1) | <0.001 |
| Free Throws Attempt | 6.6 (2.0) | 2.4 (1.9) | 1.5 (1.5) | 4.3 (2.7) | 0.9 (1.4) | 0.3 (0.8) | 2.2 (2.0) | 1.9 (1.6) | 1.1 (1.4) | <0.001 |
| Offensive Rebounds | 1.5 (1.6) | 2.1 (1.8) | 0.8 (1.0) | 2.6 (2.1) | 0.7 (0.9) | 0.2 (0.6) | 1.5 (1.4) | 1.7 (1.5) | 0.7 (0.9) | <0.001 |
| Defensive Rebounds | 3.8 (2.3) | 3.9 (2.3) | 3.1 (1.9) | 6.2 (3.1) | 1.7 (1.5) | 0.6 (0.8) | 3.2 (2.0) | 5.7 (2.6) | 2.4 (1.7) | <0.001 |
| Assists | 2.2 (1.9) | 1.5 (1.5) | 2.8 (2.2) | 2.0 (1.8) | 0.9 (1.2) | 0.3 (0.6) | 1.2 (1.3) | 1.9 (1.7) | 1.9 (1.8) | <0.001 |
| Steals | 1.6 (1.4) | 1.3 (1.3) | 1.4 (1.3) | 1.5 (1.4) | 0.6 (0.8) | 0.2 (0.5) | 0.9 (1.1) | 1.1 (1.2) | 1.0 (1.1) | <0.001 |
| Turnovers | 2.5 (1.8) | 2.2 (1.6) | 2.4 (1.7) | 2.9 (1.8) | 1.2 (1.1) | 0.5 (0.7) | 1.6 (1.3) | 2.2 (1.6) | 1.7 (1.4) | <0.001 |
| Blocks For | 0.2 (0.6) | 0.4 (0.7) | 0.2 (0.5) | 0.5 (0.8) | 0.1 (0.4) | 0.1 (0.2) | 0.3 (0.6) | 0.4 (0.7) | 0.1 (0.4) | <0.001 |
| Blocks Against | 0.3 (0.5) | 0.4 (0.7) | 0.2 (0.5) | 0.5 (0.7) | 0.1 (0.4) | 0.1 (0.2) | 0.2 (0.5) | 0.3 (0.5) | 0.2 (0.4) | <0.001 |
| Fouls Committed | 2.3 (1.2) | 2.3 (1.3) | 2.2 (1.3) | 2.3 (1.3) | 1.7 (1.3) | 0.8 (1.0) | 2.2 (1.4) | 2.3 (1.3) | 2.0 (1.3) | <0.001 |
| Fouls Received | 5.3 (1.8) | 2.7 (1.5) | 2.3 (1.5) | 4.4 (2.1) | 1.1 (1.1) | 0.3 (0.6) | 2.3 (1.5) | 2.4 (1.5) | 1.6 (1.3) | <0.001 |

Fig 5 provides a comparative visualization of observed mean values and those normalized to 40 minutes of effective playing time. Normalization produced a consistent increase across all indicators, with an average percentage gain of 23%. The most pronounced changes were observed in volume-based metrics such as field goal attempts and free throws, highlighting the influence of playing time on recorded performance.

Paired samples t-tests confirmed statistically significant differences between raw and normalized values across all performance indicators ($p < 0.05$), reinforcing the necessity of time-adjusted data when comparing players. These results emphasize that unadjusted statistics may systematically underestimate the contributions of players with reduced playing time, particularly those in supporting or rotational roles.

### 3.6. Descriptive profiles of the nine functional clusters

Figs 6–8 present both the raw and normalized descriptive values for the nine functional clusters. Time normalization revealed notable improvements in performance indicators for players within lower-participation profiles, particularly Clusters 5 and 6. Despite limited court time, these players demonstrated substantial in-game impact when their contributions were adjusted to a 40-minute standard. In contrast, players from high-participation clusters—such as Clusters 1, 3, and 4—showed more balanced distributions across performance indicators, reflecting their sustained presence and consistent involvement throughout matches.

These findings underscore the importance of using normalized data to evaluate performance in contexts where playing time varies considerably. While raw metrics may underrepresent the contributions of second-rotation players, normalization reveals their tactical relevance and efficiency when active on the court. Such patterns align with coaching practices, where strategic rotations may limit minutes despite high per-minute effectiveness.

The consistency of these profiles following normalization supports the internal validity of the segmentation. Stable specialization patterns across clusters suggest that the classification captures meaningful tactical distinctions grounded in actual competitive performance. As such, this functional typology offers a more precise and contextually relevant tool for understanding player roles in elite women's basketball, advancing traditional positional analysis by integrating time-standardized, role-specific contributions.

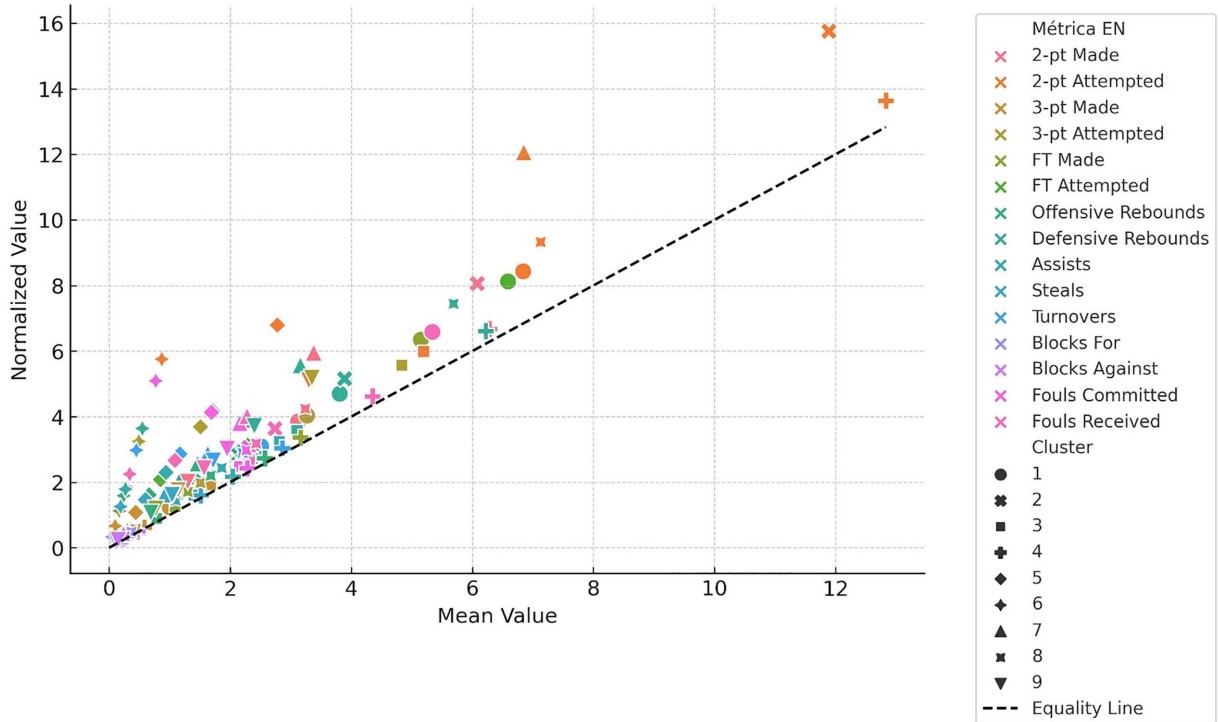

**Fig 5. Comparison between mean and normalized values of performance indicators across nine player clusters.** This scatter plot displays the relationship between raw mean values (x-axis) and normalized values (y-axis) for each performance metric, with color indicating the metric type and shape representing the player's assigned cluster (1–9). The dashed diagonal line denotes equality between the mean and normalized scales. Markers above the line indicate metrics where normalization disproportionately amplified values, while those below indicates compressed scaling.

## 4. Discussion

The primary objective of this study was to identify and describe the functional roles of players in professional women's basketball, addressing the limitations of the traditional five-position model. To achieve this, performance indicators were analyzed by conventional playing position and then restructured using a data-driven classification based on in-game behavior. While significant differences were found between traditional positions in both raw and normalized data, the cluster analysis ultimately provided greater explanatory power, revealing nine coherent functional profiles that offer a more accurate reflection of player roles within the current game context.

Normalization by effective playing time and standardization to a 40-minute reference facilitated a more precise assessment of individual impact, particularly in roles characterized by limited court time. This approach, as previously recommended [3,25,31,32] allowed for fairer comparisons between players by minimizing the confounding influence of playing time. These adjustments are especially relevant in professional women's basketball, where rotational strategies and role-specific substitutions often create disparities in participation.

Previous studies [33] has demonstrated that time-based performance metrics enhance both the interpretability of player data and the practical value of analytical models in areas such as tactical planning, talent identification, and training design. The current findings support this perspective by showing that normalized values not only change the perceived magnitude of performance but also affect how players are classified functionally. This highlights the added value of integrating context-sensitive metrics into performance analysis.

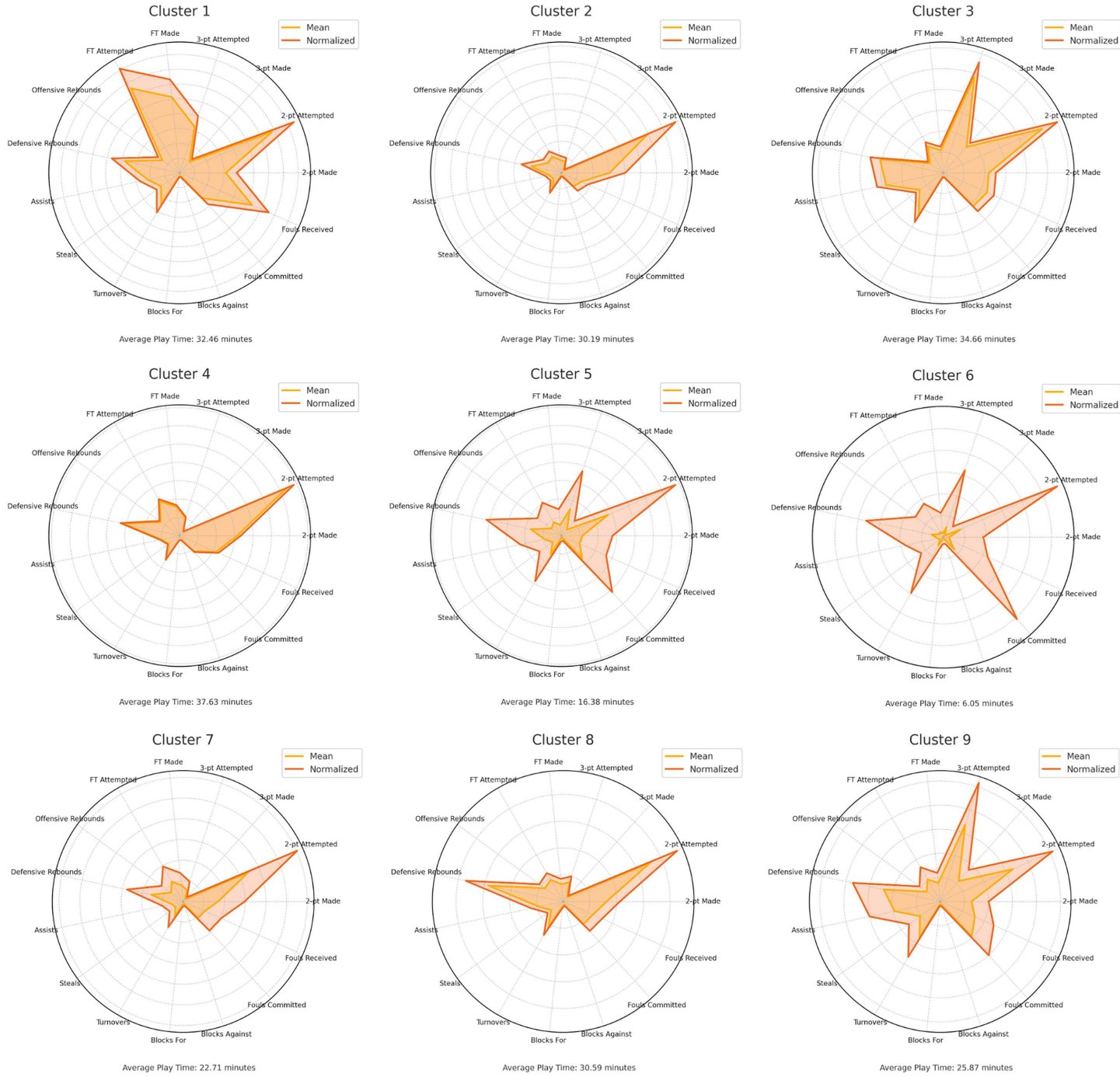

**Fig 6. Radar plots of the nine functional player clusters based on raw and normalized performance metrics.** Each panel represents a distinct cluster identified through unsupervised classification of performance indicators. Radar plots show the average raw (orange line) and normalized (red line) values for each metric, including scoring, rebounds, assists, turnovers, steals, and blocks. The average playing time for each cluster is indicated below the plots.

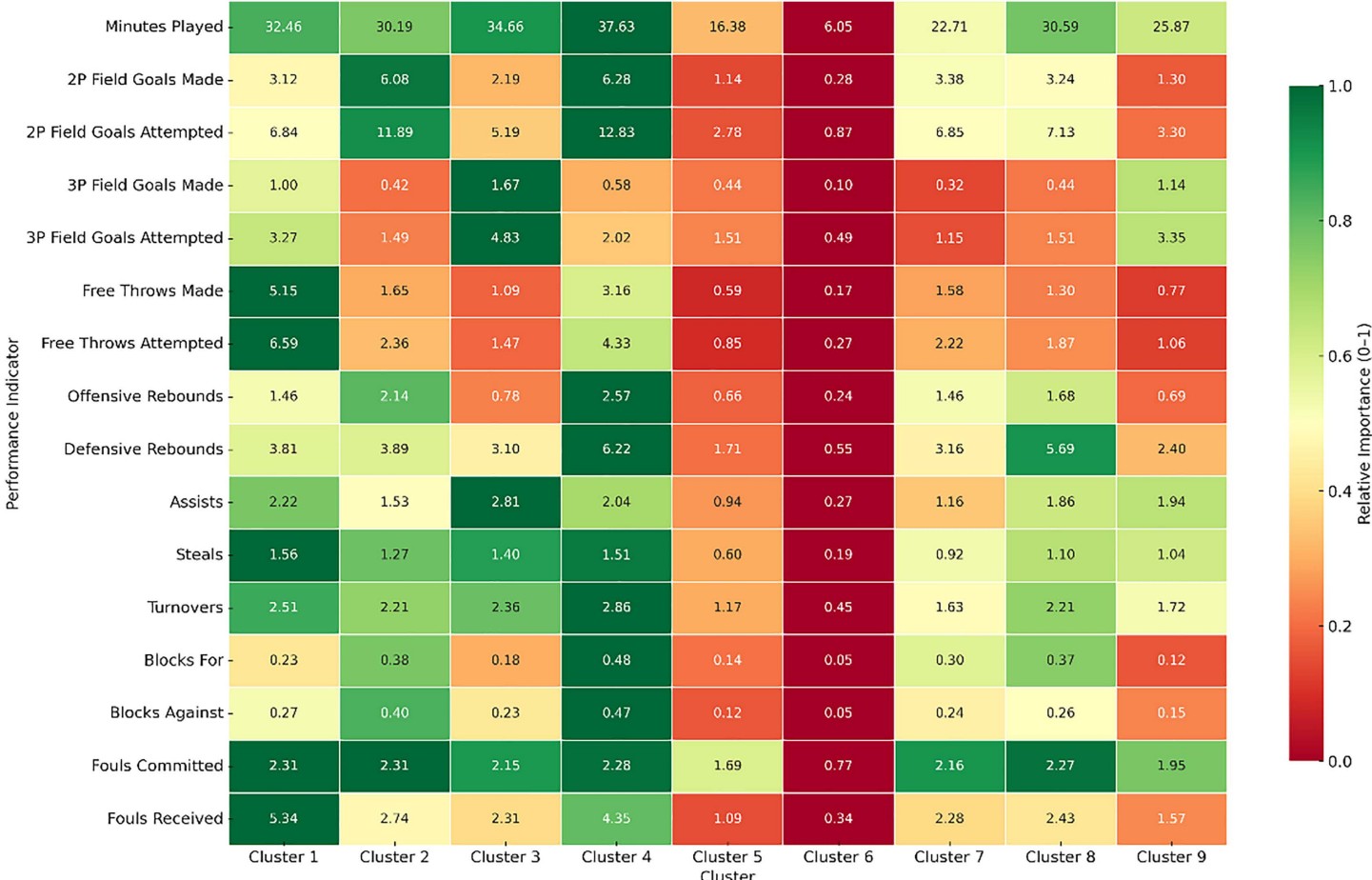

**Fig 7. Heatmap of raw players' performance classification into 9 clusters.** Color intensity represents the relative importance of each performance indicator within its row, normalized from 0 (red, lowest) to 1 (green, highest). Data are means.

The application of unsupervised clustering to box-score data—an approach established in men's basketball research—proved effective in uncovering latent structural patterns not captured through conventional positional labels. In the context of professional women's basketball, where functional classification remains underexplored [3,31], this method offers significant potential for improving both analytical insight and practical decision-making. By identifying recurring behavioral patterns in large-scale datasets, this study contributes to a more operational understanding of player functionality, informing both coaching strategies and recruitment frameworks.

## 4.1. Performance differences by traditional position

The results confirmed statistically significant differences across all performance indicators by traditional position. PGs led in assists and steals, while Cs recorded the highest values in rebounds and blocks. These findings are consistent with earlier studies [19–21], which show that traditional positions exhibit distinct statistical profiles—particularly the playmaking and perimeter shooting strengths of guards, and the rebounding dominance of interior players. Similar statistical differences across positions were also reported in previous work [17,18] where PGs showed superior shooting and Cs dominated near the basket, even within conventional classification models.

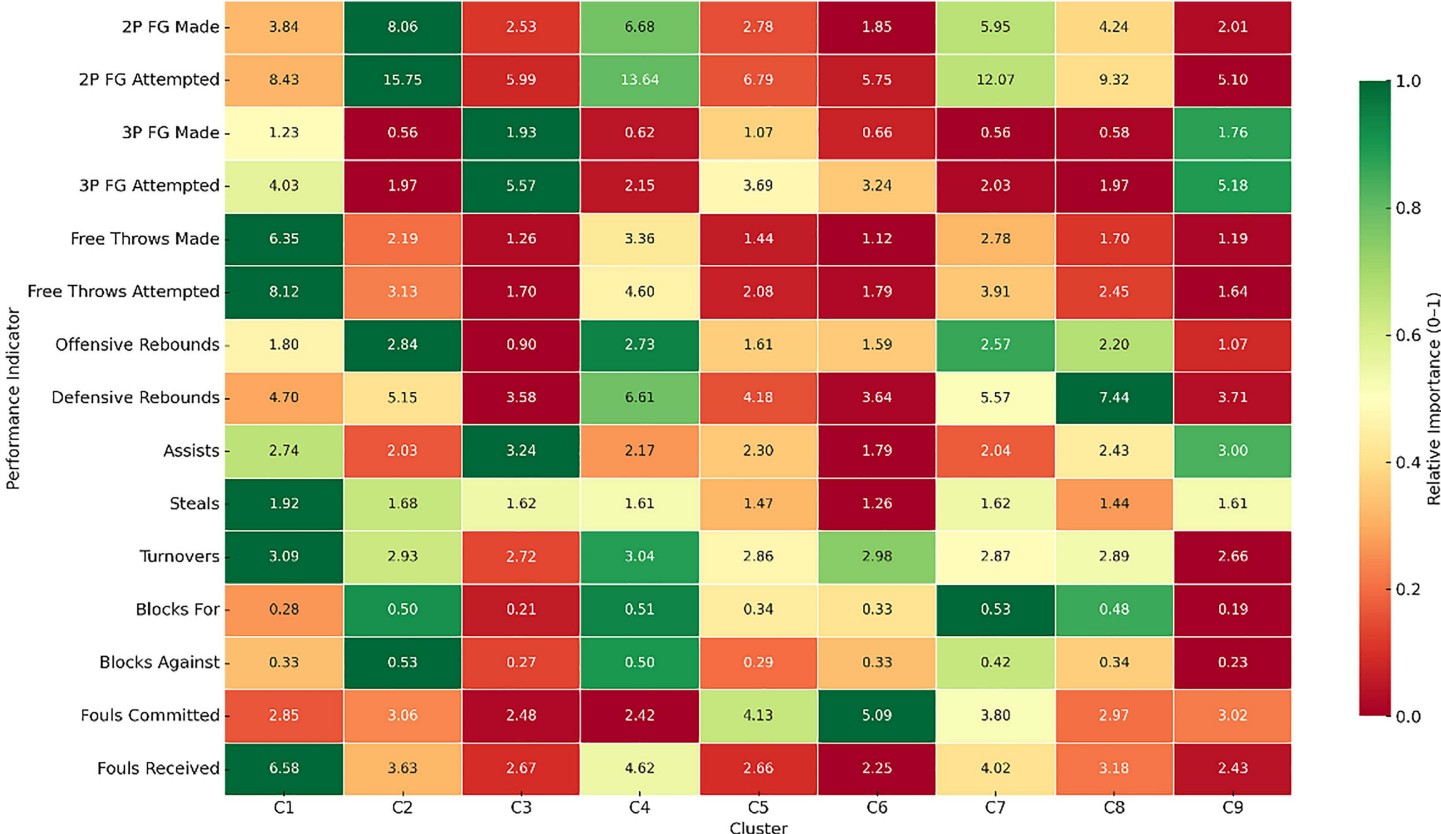

**Fig 8. Heatmap of normalized players' performance (per-40-minute) classification into 9 clusters.** Color intensity represents the relative importance of each performance indicator within its row, normalized from 0 (red, lowest) to 1 (green, highest). Data are means.

However, the functional overlaps observed between positions in this study reveal the limitations of fixed role categories. Normalized values demonstrated consistent percentage increases across all metrics, particularly among perimeter players, indicating that contributions from players with limited playing time may be systematically underestimated when using raw statistics alone. Graphical analyses confirmed these discrepancies and showed that dispersion patterns varied by action type, reinforcing the idea that positional roles are better understood when effective playing time is taken into account. For coaching staff, this adjustment is particularly valuable for accurately assessing tactical needs and optimizing lineup composition.

## 4.2. New functional classification

Segmenting players into nine clusters yielded distinct functional profiles characterized by offensive, defensive, or mixed contributions. This classification moves beyond the rigidity of the five-position system and aligns with previous findings in professional and youth basketball, where cluster-based structures have been proposed based on observed in-game performance [6,9,11,15,16]. Although these earlier studies were conducted in male populations, they consistently demonstrate that roles are more accurately derived from performance profiles than from positional labels. The present study extends this analytical framework to women's professional basketball, offering the first functional classification of its kind based on a large dataset from a top-tier competition. This approach is consistent with arguments made in recent work

[31,34], which call for redefining player roles using behavior- and efficiency-based groupings, and recognizing the transitional nature of positions across a player's career.

The observed average increase of 23% between raw and normalized values further illustrates the distorting effect of unequal playing time on perceived performance. Time-adjusted metrics revealed high-efficiency contributions from low-minute players, whose impact may otherwise be overlooked. Earlier research has highlighted the importance of using temporal normalization when comparing athletes with varying playing time [14,20,21,35], a consideration especially relevant in sports like basketball, where rotational strategies influence player exposure. Coaches frequently recognize the value of these players despite their limited minutes, focusing on efficiency over volume.

Normalizing performance metrics per minute or per possession allows for more equitable comparisons among players with differing amounts of court time. This adjustment enhances the identification of high-efficiency contributors who may have limited minutes but provide substantial impact relative to their playing time [36]. From an analytical standpoint, per-possession statistics support more robust modelling of lineup combinations based on net efficiency—defined as points scored and conceded per 100 possessions—facilitating strategic decisions that maintain team performance while managing player workload and fatigue. Incorporating tactical indicators such as ball reversal, which has been shown to promote spacing, decision-making, and offensive success in professional women's basketball, may add further value to lineup modelling and team efficiency assessments [37]. Moreover, normalized metrics help control for variations in game pace, which is particularly important when comparing players across different leagues or competitive contexts [36]. This scalability is especially useful in recruitment and scouting, where per-minute and per-possession data provide a fairer assessment of talent across diverse playing environments. Additionally, long-term analyses of elite basketball players have shown that normalized statistics (e.g., points, assists, and rebounds per minute) are effective for tracking career development and identifying key performance trends, such as increased assist rates and free-throw percentages with experience [14]. Nonetheless, not only performance but also psychosocial factors such as group cohesion can influence team sustainability. Evidence shows that stronger task cohesion among players positively predicts their intention to persist in the same team across seasons, which may contribute to long-term lineup stability and enhance the value of efficient player combinations [38].

Greater statistical variability was observed in Clusters 5 and 6, which contained players with the lowest average playing time. This instability may be explained by contextual performance factors discussed in prior analyses [39], such as match congestion, game tempo, and the high-pressure context of short substitutions. These findings underscore the need to account for both the quantity and consistency of playing time when interpreting athlete performance in team settings.

Following the methodological recommendations [31], a summary table was developed outlining the nine functional profiles, each described by tactical characteristics, representative performance indicators, and in-game roles (Table 3). The functional profiles identified offer clear tactical applications for coaching and performance planning. For example, the *All-Around High-Usage Scorer* (Cluster 4) reflects players who assume a central offensive role, combining scoring, rebounding, and foul generation—making them ideal focal points in set plays or late-game scenarios. In contrast, the *Playmaker* (Cluster 3) demonstrates high assist and three-point activity with low turnover rates, indicating a reliable perimeter organizer who facilitates offensive flow without heavy scoring reliance. The *Rim-Attacking Guard/ Foul Generator* (Cluster 1) offers value in breaking defensive structures through dribble penetration, drawing fouls, and creating mismatches. Meanwhile, *Low-Usage Defensive Anchors* (Cluster 8) contribute through paint protection and rebounding, making them essential in defensive schemes and rotations, even with minimal offensive usage. These distinctions provide coaches with actionable insights when constructing lineups, designing match-specific strategies, or developing individualized roles within the team. This framework was validated through a Delphi process involving four expert coaches with research and practical experience, ensuring both theoretical soundness and professional relevance. The resulting typology offers practical utility for tactical planning, individualized training, and scouting applications in elite women's basketball.

**Table 3. Summary table outlining the nine functional profiles, described by tactical characteristics, representative performance indicators, and in-game roles.**

| Cluster | Rotation | Court Area | Role | Description | High Performance Indicator | Low Performance Indicator |
|---|---|---|---|---|---|---|
| C1 | Starter Player | Perimeter | Slasher | Aggressive perimeter player who attacks the basket off the dribble and draws frequent fouls. Generates high free-throw volume and thrives through contact rather than shooting. | Free throws attempted and made; steals; turnovers; fouls received | Blocks in favor; fouls committed; high number of turnovers |
| C2 | Starter Player | Interior–Perimeter | Paint Scorer & Rebounder | Interior-oriented player who finishes near the rim and dominates the boards. High involvement in 2-point shooting and offensive rebounding, with limited perimeter playmaking. | Two-point shots attempted and made; offensive rebounds; blocks against | Three-point shots attempted and made; assists |
| C3 | Starter Player | Perimeter | Playmaker | Primary ball distributor with strong assist and three-point shooting rates. Drives offensive flow from the perimeter with minimal turnovers. Low impact on rebounding and defense. | Assists; three-point shots attempted and made; fewer turnovers | Offensive and defensive rebounds; blocks for and against |
| C4 | Starter Player | Perimeter–Interior | All-Around High-Usage Scorer | High-volume offensive player who combines strong inside scoring, rebounding, and foul generation. Central to team offense across multiple statistical areas. | Free throws and two-point shots attempted and made; fouls received; offensive and defensive rebounds | Fouls committed; assists |
| C5 | Second-Rotation Player | Perimeter | Efficient Support Guard | Low-turnover, role-efficient guard who contributes on both ends. Provides assists and team balance without dominating possessions. Reliable but low in scoring volume. | Fouls committed; assists; few turnovers | Three-point shots made; steals |
| C6 | Second-Rotation Player | Perimeter–Interior | Low-Usage Support Player | Plays limited minutes with minimal impact across statistics. Disciplined (few fouls or turnovers), likely used situationally or in a developmental role with focused responsibilities. | Fouls committed | Free throws and two-point shots attempted and made; assists; steals; fouls received |
| C7 | First-Rotation Player | Interior–Perimeter | Versatile interior contributor | High-work-rate interior player active in rebounding, shot contests, and drawing fouls. Impacts both ends of the floor without needing to dominate the ball. | Blocks in favor; rebounds; free throws and two-point shots; fouls received | Three-point shots attempted and made; assists |
| C8 | Starter Player | Interior | Low-Usage Defensive Anchor | Defensive specialist who protects the rim and rebounds well. Contributes little offensively but anchors the paint with physical presence and reliability. | Defensive rebounds | Three-point shots made; steals |
| C9 | First-Rotation Player | Perimeter | Three-Point Shooter | Perimeter specialist with high three-point volume and assist contribution. Offensive role is focused on spacing the floor and facilitating ball movement. | Three-point shots attempted and made; assists | Offensive rebounds; blocks for and against; turnovers; free throws and two-point shots |

### 4.3. Limitations and future directions

This study presents several limitations that should be acknowledged. The exclusive reliance on box-score statistics, while allowing for scalable and replicable analysis, does not capture contextual aspects of performance such as off-ball movement, tactical alignment, or defensive actions that are not formally recorded. These elements are critical in understanding holistic player contribution. In addition, the Delphi panel was limited to four coach-researchers. Although the panel members met strict inclusion criteria for expertise in both academic and professional domains, expanding the number of experts in future studies could enhance the robustness of cluster validation.

Future research should focus on developing multimodal performance models that integrate box-score data with tracking systems, including spatial positioning, movement intensity, and workload variables. Combining these quantitative sources with qualitative insights—such as structured interviews with coaches or expert panels—could produce more comprehensive and context-sensitive classifications. Further validation of these functional profiles across different competition levels,

such as international leagues, youth systems, or national teams, is also recommended. Longitudinal approaches would additionally support the monitoring of role evolution across a player's career, contributing to more targeted development and workload management strategies.

### 4.4. Practical applications in professional women's basketball

The results of this study offer direct applications across multiple areas of professional women's basketball, from coaching and player development to scouting and roster management. The identification of nine functional profiles—derived from players' actual statistical outputs—enables more precise strategic planning, facilitating better-informed decisions about game models and team composition.

In talent identification and recruitment, this functional classification allows clubs to target players whose profiles align with specific tactical needs. This helps move beyond reliance on traditional positional labels and supports more focused and efficient acquisition processes. Similarly, the framework aids in the definition of tactical roles that reflect each player's statistical strengths and areas for improvement, allowing for more realistic role assignments that enhance individual responsibility, tactical clarity, and overall team performance.

The classification also informs the construction of balanced lineups and rotations. By understanding the distribution of functional roles within the squad, coaching staffs can combine complementary profiles and avoid role redundancy or tactical imbalances. This optimizes team adaptability and increases the likelihood of success against varied opponents.

In the training domain, the framework enables individualized task design based on each player's functional profile. Such tailored planning supports skill acquisition and tactical development aligned with the specific demands of their in-game responsibilities, while encouraging the versatility necessary in modern high-performance environments. Simultaneously, it allows for the specialization of key roles, supporting both generalist and specialist pathways in athlete development.

Finally, this classification provides a more contextualized basis for performance evaluation. Assessing players in light of their functional role offers a fairer and more objective framework for technical assessments, minute allocation, performance monitoring, and contract negotiations. By embedding tactical function into performance metrics, coaching and management decisions gain greater relevance and accuracy.

Taken together, the findings confirm the utility of this functional classification model as a practical and strategic tool for advancing the technical, tactical, and organizational development of elite women's basketball.

## 5. Conclusions

This study demonstrates that the traditional five-position model is insufficient to describe the functional diversity observed in professional women's basketball. Through the analysis of standardized performance indicators and unsupervised clustering techniques, nine distinct functional profiles were identified and validated. This classification provides a more accurate representation of player roles, grounded in actual competitive behaviors rather than nominal position labels.

Significant differences were found across traditional positions in both raw and normalized performance metrics, reinforcing the need for refined classification systems that reflect the realities of contemporary gameplay. The proposed functional profiles capture the offensive, defensive, and hybrid specializations that define modern basketball performance.

This framework may provide a useful tool for advancing the technical, tactical, and organizational development of elite women's basketball. While the classification model shows promise in identifying functionally distinct player profiles, its application should be interpreted within the context of the dataset, which is limited to one national league and based on box-score performance metrics. Further research using broader datasets and complementary performance indicators is encouraged to validate and extend these findings.

## Acknowledgments

The authors would like to thank the expert coaches who contributed to the Delphi panel for their valuable insights and feedback during the validation of the functional profiles.

## Author contributions

**Conceptualization:** Sergio J. Ibáñez, Javier Courel-Ibáñez, José Miguel Contreras-García, María Isabel Piñar-López.

**Formal analysis:** Sergio J. Ibáñez, José Miguel Contreras-García.

**Investigation:** Sergio J. Ibáñez, Javier Courel-Ibáñez, José Miguel Contreras-García, María Isabel Piñar-López.

**Methodology:** Sergio J. Ibáñez, María Isabel Piñar-López.

**Supervision:** Sergio J. Ibáñez.

**Visualization:** Javier Courel-Ibáñez, José Miguel Contreras-García, María Isabel Piñar-López.

**Writing – original draft:** Sergio J. Ibáñez, Javier Courel-Ibáñez, José Miguel Contreras-García, María Isabel Piñar-López.

**Writing – review & editing:** Sergio J. Ibáñez, Javier Courel-Ibáñez.

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
