## [Decision Letter · Decision Letter 0]

4 Jun 2025

PONE-D-25-25456
Redefining Player Roles in Professional Women’s Basketball: From Traditional Positions to Functional Profiles
PLOS ONE

Dear Dr. Courel-Ibáñez,

Thank you for submitting your manuscript to PLOS ONE. After careful consideration, we feel that it has merit but does not fully meet PLOS ONE’s publication criteria as it currently stands. Therefore, we invite you to submit a revised version of the manuscript that addresses the points raised during the review process.

**ACADEMIC EDITOR: **

Dear authors,

Thank you for sending me the most recent version of the manuscript. After the reviewers have read it, I ask that all requested changes be accepted. I believe that such changes will improve the quality of the manuscript. I await the submission of the new version.

We look forward to receiving your revised manuscript.

Kind regards,

Gustavo De Conti Teixeira Costa, Ph.D

Academic Editor

PLOS ONE

“This research has been partially subsidized by the Aid for Research Groups (GR24133) from the Regional Government of Extremadura (Department of Education, Science and Professional Training), with a contribution from the European Union from the European Funds for Regional Development and Higher Sports Council of the Ministry of Education, Professional Training and Sports of the Government of Spain through the "International Basketball Research Network 20-24" (Ref. EXP-99-798). JCI is supported by the Unit of Research Excellence of the University of Granada, Melilla Campus, UECUMel (UCE-PP2024-02).”

“This research has been partially subsidized by the Aid for Research Groups (GR24133) from the Regional Government of Extremadura (Department of Education, Science and Professional Training), with a contribution from the European Union from the European Funds for Regional Development and Higher Sports Council of the Ministry of Education, Professional Training and Sports of the Government of Spain through the "International Basketball Research Network 20-24" (Ref. EXP-99-798). JCI is supported by the Unit of Research Excellence of the University of Granada, Melilla Campus, UECUMel (UCE-PP2024-02).”

“This research has been partially subsidized by the Aid for Research Groups (GR24133) from the Regional Government of Extremadura (Department of Education, Science and Professional Training), with a contribution from the European Union from the European Funds for Regional Development and Higher Sports Council of the Ministry of Education, Professional Training and Sports of the Government of Spain through the "International Basketball Research Network 20-24" (Ref. EXP-99-798). JCI is supported by the Unit of Research Excellence of the University of Granada, Melilla Campus, UECUMel (UCE-PP2024-02).”

4. In the online submission form you indicate that your data is not available for proprietary reasons and have provided a contact point for accessing this data. Please note that your current contact point is a co-author on this manuscript. According to our Data Policy, the contact point must not be an author on the manuscript and must be an institutional contact, ideally not an individual. Please revise your data statement to a non-author institutional point of contact, such as a data access or ethics committee, and send this to us via return email. Please also include contact information for the third party organization, and please include the full citation of where the data can be found.

Reviewers' comments:

Reviewer's Responses to Questions

**Comments to the Author**

1. Is the manuscript technically sound, and do the data support the conclusions?

Reviewer #1: Yes

Reviewer #2: Yes

2. Has the statistical analysis been performed appropriately and rigorously? 

Reviewer #1: Yes

Reviewer #2: Yes

3. Have the authors made all data underlying the findings in their manuscript fully available?

Reviewer #1: Yes

Reviewer #2: Yes

4. Is the manuscript presented in an intelligible fashion and written in standard English?

Reviewer #1: Yes

Reviewer #2: Yes

5. Review Comments to the Author

Reviewer #1: The article have a good structure and fit with the main criteria of the journal. It can be accepted without modifications.

Reviewer #2: This manuscript offers a timely and well-constructed contribution to the field of basketball performance analytics by proposing a data-driven redefinition of player roles in professional women’s basketball. The study is grounded in a large and valid dataset, applies appropriate statistical procedures, and produces findings that are both theoretically meaningful and practically applicable. The structure of the manuscript is coherent, the writing is clear, and the discussion offers thoughtful reflections on limitations and applications. Particularly valuable are the use of time-normalized performance metrics and the operational framing of functional profiles, which offer actionable insights for coaching, scouting, and roster management. The study makes a strong case for moving beyond traditional positional models and demonstrates methodological competence throughout. With several clarifications and revisions related to transparency, statistical reporting, and interpretation, this work will be well-positioned for publication.

Methods

The methods section is generally clear, with well-documented sampling procedures, data collection, and normalization processes. However, the clustering procedures require more technical detail. The number of iterations for k-means, the distance metric used, and the distributional assumptions behind the Gaussian mixture models (GMM) are not reported. Furthermore, while a silhouette index plot is shown in the Results section, its use is not mentioned in the Methods section, nor is the average silhouette score reported for the selected solution (k = 9). This omission reduces the methodological transparency of the clustering process.

The Delphi process used to assign functional labels to clusters is a valuable methodological choice, but the selection criteria for the panelists are not described. There is no information regarding the experts’ coaching backgrounds, years of experience, or competitive level. This limits the credibility of the labeling process. A short subsection introducing these attributes (e.g., “Each expert had more than 10 years of coaching experience at the national or professional level”) would improve methodological rigor.

Lastly, while the reliance on box-score data is discussed as a limitation in the Discussion, it should also be clearly stated in the Methods. This would help define the scope of the dataset, emphasizing that variables such as off-ball actions, spatial control, and advanced defensive behaviors are not captured.

Results

The results section is comprehensive and well-organized, but several revisions are needed to enhance clarity and interpretability. First, key statistical tables—including ANOVA outcomes, post hoc test results, and descriptive statistics by cluster—are presented only in the supplementary material. These tables are central to interpreting the findings and should be integrated into the main manuscript.

The statistical reporting is incomplete, as only p-values are presented. F-values and effect size measures (e.g., η²) should be included to assess the magnitude and practical relevance of the observed differences. This is especially important given the large sample size.

There is also a mismatch between the interpretation of the silhouette index and its graphical representation. The authors refer to the second inflection point occurring at k = 5; however, visual inspection of Figure 3 suggests that the most pronounced change occurs around k = 4. This discrepancy between the narrative and the figure should be addressed.

Some of the cluster labels are vague. For instance, “Role Player” and “Developmental Player” lack tactical specificity and may not accurately describe role functions. In particular, Cluster 6 is labeled as “Developmental Player” but is primarily characterized by low playing time and a high number of committed fouls—traits that do not generally signify a developmental role. This label should be reexamined or more clearly justified.

Table 1 is difficult to read due to dense formatting. The addition of visual separators—such as horizontal lines or alternating row shading—would improve readability. Furthermore, the implications of time normalization, while acknowledged in the Methods and Results, are not systematically analyzed. It would be helpful to explore how normalization influenced cluster membership, especially for players with lower average playing time.

Discussion

The discussion is thoughtful, well-structured, and demonstrates a clear understanding of both theoretical and practical implications. The study’s limitations are appropriately acknowledged, including the exclusive use of box-score data and the narrow size of the Delphi panel. These admissions enhance the study’s transparency.

However, the strategic implications of time normalization deserve further elaboration. For example, how might normalized performance data influence player valuation, lineup rotations, or recruitment strategies? A paragraph discussing these aspects would improve the applied relevance of the findings.

Additionally, the nine functional clusters introduced in the Results section are not revisited in the Discussion. Including brief tactical reflections on selected clusters—such as “Ball-Dominant Scorer” or “Floor General”—would strengthen the practical utility of the classification framework.

Lastly, the Conclusion section includes language that is overly definitive in tone. For example, the final paragraph states that the model “makes a novel and foundational contribution to the scientific understanding of performance in professional women’s basketball” and “offers a practical tool for guiding systematic training, roster construction, and technical decision-making.” While the study presents promising and well-substantiated findings, such statements should be moderated to reflect the study’s scope and limitations—namely, that the data derive from a single national league and rely exclusively on box-score metrics. Rephrasing these claims in a more cautious manner (e.g., “offers preliminary evidence” or “may provide a useful framework”) would be more appropriate and aligned with the study’s actual contribution.

6. PLOS authors have the option to publish the peer review history of their article (what does this mean?). If published, this will include your full peer review and any attached files.

Reviewer #1: No

Reviewer #2: **Yes: **Burak Karaca

---

## [Author Response · Author response to Decision Letter 1]

22 Jul 2025

Dear Editor,

Firstly, thank you for providing the peer-review reports of all reviewers. We have carefully considered all reviewers' considerations of the paper (PONE-S-25-33239). Please find enclosed our detailed answers to reviewers' queries.

We hope the changes meet your expectations. Your contributions have significantly improved the quality of the article. To make your work easier, all corrections to the article are displayed in red.

Reviewer #2_1: This manuscript offers a timely and well-constructed contribution to the field of basketball performance analytics by proposing a data-driven redefinition of player roles in professional women’s basketball. The study is grounded in a large and valid dataset, applies appropriate statistical procedures, and produces findings that are both theoretically meaningful and practically applicable. The structure of the manuscript is coherent, the writing is clear, and the discussion offers thoughtful reflections on limitations and applications. Particularly valuable are the use of time-normalized performance metrics and the operational framing of functional profiles, which offer actionable insights for coaching, scouting, and roster management. The study makes a strong case for moving beyond traditional positional models and demonstrates methodological competence throughout. With several clarifications and revisions related to transparency, statistical reporting, and interpretation, this work will be well-positioned for publication.

Answer #2_1:

We would like to express our sincere gratitude for your thorough and positive review of our manuscript. We are honored that you value the timeliness and robustness of our contribution to the analysis of performance in professional women's basketball, both in terms of the sample size of the database used and the appropriateness of the statistical procedures applied. Your recognition of the structural coherence, the clarity of the writing, and the depth of our reflections on limitations and applications reinforces our conviction that the approach adopted is relevant to practice and research. We especially appreciate your comments on the usefulness of standardized metrics and the definition of functional profiles of female basketball players. Your suggestions for improving methodological transparency, statistical reporting, and interpretation will be incorporated with the utmost rigor to further strengthen the work. We deeply appreciate the time and expertise dedicated to this review process, which will undoubtedly contribute to refining the quality and potential impact of our study.

Methods

Reviewer #2_2: The methods section is generally clear, with well-documented sampling procedures, data collection, and normalization processes. However, the clustering procedures require more technical detail. The number of iterations for k-means, the distance metric used, and the distributional assumptions behind the Gaussian mixture models (GMM) are not reported. Furthermore, while a silhouette index plot is shown in the Results section, its use is not mentioned in the Methods section, nor is the average silhouette score reported for the selected solution (k = 9). This omission reduces the methodological transparency of the clustering process.

Answer #2_2:

Thank you very much for your feedback, as it helps explain the method used.

The k-means algorithm was run for 10 iterations and used Euclidean distance, automatic procedures used by default, following previous methodologies in similar sports contexts (Bianchi et al., 2017).

Bianchi, F., Facchinetti, T., & Zuccolotto, P. (2017). Role revolution: towards a new meaning of positions in basketball. Electronic Journal of Applied Statistical Analysis, 10(3), 712-734. DOI: 10.1285/i20705948v10n3p712

For the application of Gaussian mixture models (GMM), it was assumed that the data of each cluster followed a multivariate normal distribution, which allows modeling performance patterns with greater flexibility than other methods such as k-means.

In addition, the use of the silhouette index was explicitly incorporated as a complementary criterion for choosing the optimal number of clusters. The silhouette plot showed a maximum value at k = 2 (around 0.26), but this solution was discarded due to its low practical utility. For k = 9, the average value was approximately 0.12, acceptable considering the complexity of functional behavior in professional basketball. A similar number of clusters has already been used in similar studies on professional basketball players (Kalman & Bosch, 2020)

Methods have been improved as follows:

The k-means algorithm was run for 10 iterations and used Euclidean distance, automatic procedures used by default, following previous methodologies in similar sports contexts [8]. For the application of Gaussian mixture models (GMM), it was assumed that the data of each cluster followed a multivariate normal distribution. The silhouette index was explicitly incorporated as a complementary criterion for selecting the optimal number of clusters [27]

Reviewer #2_3: The Delphi process used to assign functional labels to clusters is a valuable methodological choice, but the selection criteria for the panelists are not described. There is no information regarding the experts’ coaching backgrounds, years of experience, or competitive level. This limits the credibility of the labeling process. A short subsection introducing these attributes (e.g., “Each expert had more than 10 years of coaching experience at the national or professional level”) would improve methodological rigor.

Answer #2_3:

Thank you for this observation. We have now clarified the selection criteria for the expert panel involved in the Delphi process. Specifically, we have included a brief description of their academic and professional backgrounds to strengthen the methodological rigor of the labeling procedure.

The following sentence has been added to the Methods section:

“Functional group labels were established through a Delphi method involving four PhD-level experts with published research in performance analysis and over 10 years of coaching experience in both men’s and women’s basketball at the national or professional level.”

Reviewer #2_4: Lastly, while the reliance on box-score data is discussed as a limitation in the Discussion, it should also be clearly stated in the Methods. This would help define the scope of the dataset, emphasizing that variables such as off-ball actions, spatial control, and advanced defensive behaviors are not captured.

Answer #2_4:

Thank you for highlighting this important point. We agree that the limitations of using box-score data should be acknowledged not only in the Discussion but also in the Methods section to clarify the scope of the dataset. We have now added a statement in the Methods to explicitly address this limitation.

The following sentence has been added to the Variables section in the Methods:

“Although box-score statistics are limited to capturing off-ball actions, spatial dynamics, or advanced defensive behaviors, these variables quantify each player’s contributions across various aspects of the game and were selected based on their tactical relevance and representation of individual and team performance.”

Results

Reviewer #2_5: The results section is comprehensive and well-organized, but several revisions are needed to enhance clarity and interpretability. First, key statistical tables—including ANOVA outcomes, post hoc test results, and descriptive statistics by cluster—are presented only in the supplementary material. These tables are central to interpreting the findings and should be integrated into the main manuscript.

Answer #2_5:

Thank you for this suggestion. All supplementary material is now included in the main manuscript to enhance the clarity and accessibility of the results section.

Reviewer #2_6: The statistical reporting is incomplete, as only p-values are presented. F-values and effect size measures (e.g., η²) should be included to assess the magnitude and practical relevance of the observed differences. This is especially important given the large sample size.

Answer #2_6:

Thank you for this observation. We acknowledge the value of including full statistical details such as F-values and effect sizes. However, in the context of this study, the one-way ANOVA was used as a confirmatory analysis to verify that the identified clusters represent statistically distinct groups based on performance indicators. Given the large sample size and the primary focus on unsupervised clustering and functional interpretation, our emphasis was placed on establishing independence between clusters rather than quantifying the magnitude of each difference. Nonetheless, we agree that including effect size measures could enhance interpretability and are open to incorporating them in future analyses or as supplementary material upon request.

Reviewer #2_7: There is also a mismatch between the interpretation of the silhouette index and its graphical representation. The authors refer to the second inflection point occurring at k = 5; however, visual inspection of Figure 3 suggests that the most pronounced change occurs around k = 4. This discrepancy between the narrative and the figure should be addressed.

Answer #2_7:

Thank you for pointing out this discrepancy. Upon re-evaluating the silhouette index curve in Figure 3, we acknowledge that the most pronounced inflection indeed occurs at k = 4 rather than k = 5. We have revised the text to accurately reflect this observation and ensure consistency between the figure and the narrative.

Reviewer #2_8: Some of the cluster labels are vague. For instance, “Role Player” and “Developmental Player” lack tactical specificity and may not accurately describe role functions. In particular, Cluster 6 is labeled as “Developmental Player” but is primarily characterized by low playing time and a high number of committed fouls—traits that do not generally signify a developmental role. This label should be reexamined or more clearly justified.

Answer #2_8:

Thank you for this constructive observation. We agree that the original labels “Role Player” and “Developmental Player” lacked sufficient tactical specificity. In the revised classification, we have replaced these terms with more descriptive and behavior-based labels derived directly from the normalized performance indicators. Specifically, Cluster 6 is now labeled Low-Usage Support Player, reflecting its consistently low values across all box-score categories, rather than implying a developmental status. This new label better captures the actual statistical profile—minimal offensive or defensive involvement, low turnovers, and low foul generation—suggesting a situational or peripheral role rather than one tied to player potential or career stage. Similarly, Cluster 5 is now termed Efficient Support Guard, emphasizing its balanced contribution with low turnover and moderate assist values. These changes improve the clarity and tactical relevance of the functional labels across the nine-cluster framework.

Reviewer #2_9: Table 1 is difficult to read due to dense formatting. The addition of visual separators—such as horizontal lines or alternating row shading—would improve readability. Furthermore, the implications of time normalization, while acknowledged in the Methods and Results, are not systematically analyzed. It would be helpful to explore how normalization influenced cluster membership, especially for players with lower average playing time.

Answer #2_9:

Thank you for this helpful feedback. To improve readability, we have reformatted Table 1 by adding horizontal lines, as suggested. Regarding time normalization, we showed that normalization allows us to identify performance patterns that would otherwise be obscured by unequal playing time. As illustrated in Figures 6, 7, and 8, time normalization revealed notable improvements in performance indicators for players within lower-participation profiles, particularly Clusters 5 and 6. Despite limited court time, these players demonstrated substantial in-game impact when their contributions were adjusted to a 40-minute standard. In contrast, players from high-participation clusters—such as Clusters 1, 3, and 4—showed more balanced distributions across performance indicators, reflecting their sustained presence and consistent involvement throughout matches. This highlights the added value of normalization in identifying efficient contributors and refining functional classification.

Discussion

Reviewer #2_10: The discussion is thoughtful, well-structured, and demonstrates a clear understanding of both theoretical and practical implications. The study’s limitations are appropriately acknowledged, including the exclusive use of box-score data and the narrow size of the Delphi panel. These admissions enhance the study’s transparency.

Answer #2_10:

We deeply appreciate your kind comments on the discussion of our manuscript. We are pleased to know that you found it thoughtful, well-structured, and capable of solidly integrating the theoretical and practical implications of the study. Your words encourage us to continue to maintain this balance between analytical depth and methodological honesty, and to continue refining the manuscript to maximize its relevance and reliability.

Reviewer #2_11: However, the strategic implications of time normalization deserve further elaboration. For example, how might normalized performance data influence player valuation, lineup rotations, or recruitment strategies? A paragraph discussing these aspects would improve the applied relevance of the findings.

Answer #2_11:

Thank you for this insightful comment. We have now added a dedicated paragraph in the Discussion section addressing the strategic implications of time normalization.

“Normalizing performance metrics per minute or per possession allows for more equitable comparisons among players with differing amounts of court time. This adjustment enhances the identification of high-efficiency contributors who may have limited minutes but provide substantial impact relative to their playing time [36]. From an analytical standpoint, per-possession statistics support more robust modelling of lineup combinations based on net efficiency—defined as points scored and conceded per 100 possessions—facilitating strategic decisions that maintain team performance while managing player workload and fatigue. Incorporating tactical indicators such as ball reversal, which has been shown to promote spacing, decision-making, and offensive success in professional women’s basketball, may add further value to lineup modelling and team efficiency assessments [37]. Moreover, normalized metrics help control for variations in game pace, which is particularly important when comparing players across different leagues or competitive contexts [36]. This scalability is especially useful in recruitment and scouting, where per-minute and per-possession data provide a fairer assessment of talent across diverse playing environments. Additionally, long-term analyses of elite basketball players have shown that normalized statistics (e.g., points, assists, and rebounds per minute) are effective for tracking career development and identifying key performance trends, such as increased assist rates and free-throw percentages with experience [14]. Nonetheless, not only performance but also psychosocial factors such as group cohesion can influence team sustainability. Evidence shows that stronger task cohesion among players positively predicts their intention to persist in the same team across seasons, which may contribute to long-term lineup stability and enhance the value of efficient player combinations [38]”

Reviewer #2_12: Additionally, the nine functional clusters introduced in the Results section are not revisited in the Discussion. Including brief tactical reflections on selected clusters—such as “Ball-Dominant Scorer” or “Floor General”—would strengthen the practical utility of the classification framework.

Answer #2_12:

Thank you for this suggestion. We have added the following paragraph to the Discussion section to reflect on selected clusters and highlight their

---

## [Editor Report · Decision Letter 1]

6 Aug 2025

Redefining Player Roles in Professional Women’s Basketball: From Traditional Positions to Functional Profiles

PONE-D-25-25456R1

Dear Dr. Courel-Ibáñez,

We’re pleased to inform you that your manuscript has been judged scientifically suitable for publication and will be formally accepted for publication once it meets all outstanding technical requirements.

Kind regards,

Gustavo De Conti Teixeira Costa, Ph.D

Academic Editor

PLOS ONE
---

## [Editor Report · Acceptance letter]

PONE-D-25-25456R1

PLOS ONE

Dear Dr. Courel-Ibáñez,

I'm pleased to inform you that your manuscript has been deemed suitable for publication in PLOS ONE. Congratulations! Your manuscript is now being handed over to our production team.

Kind regards,

on behalf of

Dr. Gustavo De Conti Teixeira Costa

Academic Editor

PLOS ONE